# Biosynthesis of Bacterial Nanocellulose from Low-Cost Cellulosic Feedstocks: Effect of Microbial Producer

**DOI:** 10.3390/ijms241814401

**Published:** 2023-09-21

**Authors:** Ekaterina A. Skiba, Nadezhda A. Shavyrkina, Maria A. Skiba, Galina F. Mironova, Vera V. Budaeva

**Affiliations:** 1Laboratory of Bioconversion, Institute for Problems of Chemical and Energetic Technologies, Siberian Branch of the Russian Academy of Sciences (IPCET SB RAS), 659322 Biysk, Russia; 32nadina@mail.ru (N.A.S.); yur_galina@mail.ru (G.F.M.); 2Higher Chemical College of the Russian Academy of Sciences, Mendeleev University of Chemical Technology of Russia, 9, Miusskaya Square, 125047 Moscow, Russia; skibamaria21@gmail.com

**Keywords:** bacterial cellulose, miscanthus, oat hulls, *Komagataeibacter xylinus*, *Medusomyces gisevii*, Kombucha, SCOBY, crystallinity index

## Abstract

Biodegradable bacterial nanocellulose (BNC) is a highly in-demand but expensive polymer, and the reduction of its production cost is an important task. The present study aimed to biosynthesize BNC on biologically high-quality hydrolyzate media prepared from miscanthus and oat hulls, and to explore the properties of the resultant BNC depending on the microbial producer used. In this study, three microbial producers were utilized for the biosynthesis of BNC: individual strains *Komagataeibacter xylinus* B-12429 and *Komagataeibacter xylinus* B-12431, and symbiotic *Medusomyces gisevii* Sa-12. The use of symbiotic *Medusomyces gisevii* Sa-12 was found to have technological benefits: nutrient media require no mineral salts or growth factors, and pasteurization is sufficient for the nutrient medium instead of sterilization. The yield of BNCs produced by the symbiotic culture turned out to be 44–65% higher than that for the individual strains. The physicochemical properties of BNC, such as nanofibril width, degree of polymerization, elastic modulus, Iα allomorph content and crystallinity index, are most notably dependent on the microbial producer type rather than the nutrient medium composition. This is the first study in which we investigated the biosynthesis of BNC on hydrolyzate media prepared from miscanthus and oat hulls under the same conditions but using different microbial producers, and showed that it is advisable to use the symbiotic culture. The choice of a microbial producer is grounded on the yield, production process simplification and properties. The BNC production from technical raw materials would cover considerable demands of BNC for technical purposes without competing with food resources.

## 1. Introduction

Due to the unfavorable environmental situation on the planet, there is much concern about the transition to a circular economy. Biodegradable polymers are undoubtedly a sound answer to the global ecological challenge because they are not only capable of replacing synthetic polymers, but also open up opportunities to create conceptually new materials [1,2]. One example of such materials is bacterial nanocellulose (BNC), a bacteria-synthesized polymer. The research on BNC has considerably elaborated the insights into classical materials science, owing to the nanodimension of this material and some of the properties this nanodimension provides [3,4]. The application range of BNC is immensely wide, starting from foods and biomedical devices to electronics and energy production [4,5]. This is explained by the fact that BNC, besides its native state, can also be used as a part of composites [6,7,8]. That said, BNC can act as both a matrix and a reinforcement, and it can be modified both in situ and ex situ [9,10]. It is expected that the technical field of BNC application will hold a leading position in the nearest future [3,5].

The wide adoption of BNC is impeded by its high prime cost, with the cost of a nutrient medium constituting ~30% of the total production cost of BNC [11]. Among the latest trends is the concept of the production of valuable BNC from low-cost plant cellulose. This idea has become so globally recognized that not only experimental research papers, but also numerous reviews are devoted to it [12,13,14,15,16,17,18,19,20,21,22]. At the same time, despite the economic attractiveness, the problem of BNC production from plant cellulose is very technically difficult and therefore remains unresolved so far.

In BNC production, the fermentation problem compounds the problems related to cellulosic feedstock pretreatment and substrate fermentability, which are common for the use of lignocellulose, because the composition of nutrient media required by microbial producers of BNC is very specific [13,15]. We have previously demonstrated a successful biosynthesis of BNC in nutrient media derived from oat hulls and miscanthus. Oat hulls are a feedstock with a zero prime cost, since all the expenses are already allocated to cereal production [23]. Miscanthus has been recognized as one of the leading sources of perennial biomass for the production of high-value-added products [24]. Both of the feedstocks are globally abundant.

The process stages involved the chemical pretreatment of feedstocks with dilute HNO_3_ and/or NaOH solutions and the enzymatic hydrolysis of the solid residue from the chemical pretreatment. The two-stage pretreatment of feedstocks with dilute HNO_3_ and NaOH was shown to afford a biologically good nutrient medium, in which case the *Medusomyces gisevii* Sa-12 culture was capable of synthesizing BNC with standard structural characteristics, irrespective of the feedstock type and its pretreatment method: a 86–93% crystallinity index and a 96–98% Iα allomorph content [25,26].

However, the issue regarding the use of the symbiotic culture caused substantial disputes among our reviewers and necessitated the continuation of our research. It is clear that the microbiota composition varies according to the culturing conditions and geographical point at which the symbiotic culture develops [27,28]; that the use of the consortium leads to an improper utilization of the substrate to form multiple metabolites of the symbiotic microorganisms; that the productivity of the consortium towards BNC is instable; and that the in-process control is extremely sophisticated and cannot be automated.

Interestingly, the symbiotic culture Kombucha is the central natural object for the isolation of pure cultures of high-productivity microbial producers of BNC, the acetobacteria of the genus *Komagataeibacter* [29]. Therefore, the choice between the consortium and individual strains derived therein is becoming an increasingly debated topic in actual production.

In this study, we investigated the biosynthesis of BNC on good-quality nutrient media prepared from miscanthus and oat hulls and tried out both the individual strains and the symbiotic culture as microbial producers. The present study aimed to answer the question of which microbial producer is most suitable for biosynthesizing BNC from low-cost cellulosic feedstocks to achieve a high yield of BNC, and how the choice of microbial producer affects the properties of the resulting macromolecules in this case. The answer to that question holds significant importance for designing a commercial production of high-value BNC from low-cost cellulosic feedstocks.

## 2. Results

### 2.1. Biosynthesis of BNC on Hydrolyzate Media

The hydrolyzate media were prepared from miscanthus and oat hulls under the same conditions. The first stage involved chemical pretreatment of the feedstocks, followed by enzymatic hydrolysis of the resultant pulps [25]. The media were standardized against the content of reducing sugars (RS), and the biosynthesis of BNC was conducted using three microbial producers under the same conditions.

The time profiles of the RS concentration during the biosynthesis of BNC are depicted in Figure 1. The major loss of RS was observed in the first 7 days; afterwards, the curves reached a plateau. The residual RS concentrations on the hydrolyzate media were 7 g/L for the B-12429 strain, 4–5 g/L for the B-12431 strain, and 5.5–6.5 g/L for the symbiotic culture. This was the maximum concentration, while 0.5% was the minimum concentration on the control medium.

The time courses of pH are displayed in Figure 2. The peculiar feature of the B-12429 strain was the minimum pH of 3.5 observed at 4 days of biosynthesis; afterwards, the pH gradually increased to 3.9–4.3. The pH declined to 3.6–3.9 at 3–5 days of biosynthesis for the B-12431 strain and the symbiotic culture and remained unchanged until the end of the experiment. Minimum pH values were recorded for the symbiotic culture: about 3.5 on the hydrolyzate media and about 3.0 on the control nutrient medium.

The curves of the acetobacterial count are shown in Figure 3. It is seen from the graphs that no lag phase was observed in the cultivation of the individual strains, which is likely due to pre-adaptation of the producers to the medium used. Between 2 and 14 days, a stationary phase was observed. The bacterial count at 7 days when bacteria were cultured on the hydrolyzates was 3–4 times as high as that of bacteria cultured in the control medium. The bacterial count for the B-12431 strain on all the nutrient media was 1–3% higher than that for B-12429, which is insignificant. The lag phase was observed in the cultivation of the symbiotic culture on all the media used, which is explained by the metabolic peculiarities of the consortium: the yeast proliferated first, and the metabolic products of yeast were then utilized as a substrate by the acetobacteria, which began to increase in number. The acetobacterial count is known to serve as a marker of BNC synthesis; that is, the higher the bacterial concentration, the higher the BNC yield [30]. One could predict the maximum yield of BNC on the oat hull-derived medium for the individual strains and on the control medium for the symbiotic culture.

The highest yield of BNC when the individual strains were used was achieved on the oat-hull hydrolyzate medium (Figure 4): 3.5% for B-12429 and 4.1% for B-12429. Interestingly, the BNC yield in the experiment with B-12429 on the miscanthus-derived medium was close to that on the oat hull-derived medium, whereas the experiment with B-12431 showed a large gap between the yield curves. The biosynthesis of BNC in Hestrin–Schramm media took place in a different way as well: while the BNC yield kept growing for up to 14 days in the experiment with B-12429, it reached a plateau at as early as 7 days in the experiment with B-12431.

The lowest yields of BNC were obtained with the individual *Komagataeibacter xylinus* strains on control medium and were 76–78% as low as that obtained with *Medusomyces gisevii* Sa-12 on the control medium (2.2–2.4% vs. 10%). That said, the yields we obtained on the control medium are in complete agreement with those reported in the literature for these microbial producers, i.e., for the B-12429 strain [31] and for the B-12431 strain [32].

### 2.2. Physicochemical Characterization of BNC Test Samples

Standard analytical methods commonly used in research worldwide were employed to perform the physicochemical characterization of BNC [3,4,13]. The bacterial origin of BNC is evidenced by its nanodimension, which is confirmed via scanning electron microscopy (SEM) [3]. SEM images of the BNC samples produced by the individual strains and symbiotic culture on the hydrolyzate and control media are depicted in Figure 5. Regardless of the pretreatment method for obtaining a substrate, all the BNC samples obtained on miscanthus- and oat hull-derived hydrolyzates were found to exhibit an unordered reticulate structure, the fingerprint of BNC [3,4]. The width of microfibrils based on the SEM images (calculation was performed using the ImageJ image processing program) were measured to be from 82 nm to 91 nm; no significant difference in the microfibril width was detected.

Chemical purity is the second important fingerprint of BNC that differentiates it from the other types of cellulose [3]. Microorganisms synthesize BNC devoid of hemicelluloses and lignin, and washing helps purify BNC from nutrient medium residues and microbial cells.

The absorption band assignments for functional groups of the BNC samples obtained from miscanthus and oat hulls using the *Komagataeibacter xylinus* B-12429 and B-12431 strains are given in Appendix A (see the Appendix A for this article). All the BNC samples exhibit close values of stretching vibrations matching those of cellulose. Specifically, absorption bands at 3434 cm^−1^ are due to the OH-group stretching, and absorption bands at 2894 cm^−1^ are due to the CH_2_ and CH stretching. Absorption bands at 1634 cm^−1^ refer to the bending vibrations of OH groups tightly bound to water. Absorption bands near 1432–1369 cm^−1^ point to CH_2_ and CH bending. Absorption bands near 1318 cm^−1^ indicate bending vibrations of the primary alcohol OH group, those near 1280 cm^−1^ indicate the alcohol CH_2_ group bending, and those near 1163 cm^−1^ indicate the bending vibrations of the alcohol C-O-C and C-O groups. Absorption bands near 896 cm^−1^ corroborate the presence of β-1,4 glycosidic bonds between the glucose molecules. Thus, the infrared spectroscopy results confirmed that the cellulose samples had chemical functional abilities to react with radicals and cations of metals, holding them on the surface [33].

Any type of cellulose can be characterized by a universal measure such as the degree of polymerization (DP) (Table 1). The DP of BNC obtained with *Komagataeibacter xylinus* B-12429 was 800–900, which is 1.8–2.9 times lower than that of BNC obtained with *Medusomyces gisevii* Sa-12 (1450–2600). The BNC synthesized using *Komagataeibacter xylinus* B-12431 had a DP of 1900–2400, which is almost the same as that of BNC obtained with *Medusomyces gisevii* Sa-12. The lowest DP of BNC produced by *Komagataeibacter xylinus* B-12429 was observed when grown on the miscanthus hydrolyzate, while for *Komagataeibacter xylinus* B-12431, the lowest DP of BNC was observed when grown on the control medium.

The elastic moduli of BNCs obtained using the individual strains were higher than those of BNCs obtained with the symbiotic culture, i.e., by 1.1–2.4 times for B-12429 (1033–2112 MPa vs. 510–933 MPa) and 2.2–4.3 times for B-12431 (2215–2243 MPa vs. 510–933 MPa).

The onset temperatures of pyrolysis for all the samples were similar, irrespective of the medium in which the samples were obtained, and ranged from 290.7 °C to 329.3 °C, with this value for the symbiotic culture being slightly higher than that for the individual strains, but the difference cannot be considered significant.

The analysis of X-ray patterns is detailed in [34] for BNCs obtained using the individual strains and in [35,36] for BNCs obtained with the symbiotic culture. The estimation results for X-ray patterns via full-profile analysis showed that the structure of all the samples 92.8–100.0% matched Iα allomorph. High crystallinity indices were achieved, which depended on the strain used rather than the cellulosic feedstock type (miscanthus or oat hulls): the crystallinity index varied from 86% to 94% for *Medusomyces gisevii* Sa-12, from 74% to 90% for *Komagataeibacter xylinus* B-12429, and from 89% to 100% for *Komagataeibacter xylinus* B-12431.

## 3. Discussion

### 3.1. BNC Biosynthesis Process on Hydrolyzate Media

The issue of whether to use individual strains or a symbiotic culture for the biosynthesis of BNC has become particularly acute in the literature worldwide over the last five years [37,38,39]. The use of individual strains has a range of undisputed advantages: it not only offers high productivity with respect to the target product (BNC) and prospects for a further productivity enhancement via gene engineering techniques, but also the predictability of metabolism and the possibility of controlling the population count and condition [13,18,40]. At the same time, the poor stability of individual strains under production conditions (a spontaneous decline or a complete loss in the cellulose-synthesizing capability of microbial producers has been documented multiple times) impels researchers to look for new producers. One method involves stabilizing the functioning of strains using gene engineering techniques [12,15,40], and the other one involves utilizing consortia [37,41].

The “tea fungus” consortium was described in 220 B.C. [30], and is distributed over a wide geographical range under many names: it is called Manchurian mushroom, Kombucha, tea Kvass, starter tea, Japanese mushroom, and Russian mushroom in the European countries [27]; this microbial producer is called Hamza’s Khubdat in the Arabian countries and Haipao in Taiwan [42]; it is called SCOBY (symbiotic culture of bacteria and yeast) in Korea [43]. Herein, we call this symbiotic culture *Medusomyces gisevii*, a formal systematic name given to the consortium by Lindau [44]. These many names evidence that the consortium is very popular and highly adaptive.

*Medusomyces gisevii* is a natural, extremely conservative and, at the same time, extremely stable consortium. The consortia have a high adaptive potential, and the coordinated consumption of a substrate and the formation of metabolites enhance their tolerance to nutrient media of a variable composition and increase the target product (BNC) yield, which is of special importance in a real production setting [45,46]. Therefore, the whole trend of using the symbiotic culture for BNC biosynthesis has emerged in the literature worldwide in the recent times [33,38,39,41,47,48,49,50,51,52,53,54].

From a technological standpoint, it is extremely interesting that non-sterile nutrient media can be used to culture *Medusomyces gisevii* [43,45]. The use of black tea extract makes it unnecessary to add nutrient salts and vitamins, which are usually utilized for BNC biosynthesis [37].

Based on the present study, Table 2 compares the organization difficulties and efficiencies of BNC biosynthesis on hydrolyzate media using individual strains and the symbiotic culture and suggests that the use of the symbiotic culture has more pros than cons.

When the individual strains were cultured on hydrolyzate media, the BNC yield increased by 38–46% for B-12429 and by 150–242% for B-12431 compared to the synthetic medium. The phenomenon of the higher BNC yield on hydrolyzate media compared to the synthetic one has previously been for individual strains on many occasions [13,19]. It was hypothesized that it could be due to the hydrolyzates containing soluble cello-oligosaccharides that are utilizable by strains that contain endo-1,4-β-glucanase [55].

Herein, the biosynthesis of BNC using individual strains and a symbiotic culture was examined for the first time under identical conditions on identical hydrolyzate media. Despite the increased BNC yield obtained with the individual *Komagataeibacter xylinus* strains on hydrolyzate media compared to the control medium, this BNC yield was found to be 59–65% and 44–48% lower than that obtained with the symbiotic culture on the oat hull-derived medium and on the miscanthus-derived medium, respectively. This proves that miscanthus itself contains inhibitors of BNC biosynthesis, as the process stages were performed in the same way.

It can be inferred that the individual strains cannot compete with the symbiotic culture when cultivated on enzymatic hydrolyzates prepared from easily renewable cellulosic biomass. Apart from the critically low BNC yield, the individual strains require that (1) the medium be supplemented with nutrients without which BNC almost stops growing and that (2) nutrient media be sterilized, whereas the symbiotic culture suffices for pasteurization at 100 °C. These two factors are essential technological benefits in the commercial production of BNC [13].

The conversion of a pristine cellulosic feedstock into BNC is a very complicated process whose efficiency is influenced by multiple factors. When it comes to BNC production from cellulosic raw materials using Kombucha, our research stands alone, and we have no comparative studies to reference. Therefore, we reviewed studies in which individual strains were utilized for BNC production from cellulosic feedstock. For instance, Cheng et al. [56] used corn stalks as the feedstock, performed hydrolysis with acetic acid, and synthesized BNC for 7 days using *Acetobacter xylinum* ATCC 23767, with the BNC volumetric productivity being 2.93 g/L and the yield being 9.22%. Vasconcellos et al. [57] also utilized corn stalks as the feedstock subjected to hydrothermal treatment, and synthesized BNC for 7 days by *Komagataeibacter hansenii* ATCC 23769, with the BNC volumetric productivity being 0.71 g/L. Goelzer et al. [58] used rice husk, which was subjected to enzymatic hydrolysis, and synthesized BNC for 10 days using *Acetobacter xylinum* ATCC 23769, with the BNC volumetric productivity being 1.6–2.4 g/L and the yield being 4.4–6.7%. These three studies align well with the findings obtained herein.

Son et al. [24] utilized miscanthus as the feedstock, pretreated it with sulfuric acid and synthesized BNC for 4 days using *Gluconacetobacter xylinus* ATCC 53524, with the BNC volumetric productivity being 14.9 g/L. However, the synthesized product was not BNC, in fact, but a co-polymer of BNC and sodium alginate; therefore, these results cannot be compared. We are not aware of other examples of the deep transformation of miscanthus into BNC.

The literature reports extremely high yields of BNC resulting from the conversion of lignocellulosic biomass. Chen et al. [59] used wheat straw as the feedstock treated with ionic liquid followed by enzymatic hydrolysis, and synthesized BNC for 7 days using *Gluconacetobacter xylinus* ATCC 23770, with the BNC volumetric productivity being 8.3 g/L. Hong et al. [55] utilized olden uncolored T-shirts, which were treated with ionic liquid and then enzymatically hydrolyzed, and synthesized BNC for 14 days by *Gluconacetobacter xylinus* ATCC 23770, with the volumetric productivity being 10.8 g/L. These results are 4.6–6.0 times better than those achieved in the present study. However, the commercial processing of lignocellulosic biomass does not currently use ionic liquids. Moreover, the commercial production of BNC using individual strains is quite limited [59], whereas the commercial production of BNC using a symbiotic culture is growing steadily worldwide, including the use of BNC for technical purposes [48,59]. The result obtained in the present study is world-class.

### 3.2. Physicochemical Properties of BNC Test Samples

All the studied BNC samples, irrespective of the microbial producer and nutrient medium, have a 3D reticulate structure typical of BNC in particular [4], as determined by SEM, and are chemically pure celluloses, as determined by IR spectroscopy, which distinguishes BNC from the other cellulose types [33].

Herein, the elastic modulus and the degree of polymerization of the BNC samples depended mostly on the microbial producer type and slightly on the source of the glucose nutrient medium. In the literature, the BNC elastic modulus varies greatly from 15–138 GPa [15,40] to 10–17 MPa [60]. The elastic modulus of BNC obtained using Kombucha was reported to be 32 MPa [50]. Thus, the elastic moduli achieved herein are in good agreement with the literature data.

The degree of polymerization (DP) is a universal measure for any cellulose type, but BNC stands out among other celluloses for a higher DP [3,4,40]. The DPs achieved herein for BNCs are on a par with the literature values [40,50]. The literature rarely discusses the variations in the DP during the biosynthesis. All the microbial producers examined herein showed a minor decline in the DP after 14 days compared to 7 days, which is strongly consistent with the literature data. The same tendency was described in [61], where this fact was difficult to explain, and in [62], where it was explained by the action of cellulases being produced by the *A. xylinum* used. It could be more logical to suggest that physiology lies behind this tendency. Acetic acid bacteria are known to be strict aerobes; therefore, biosynthesis of new BNC layers occurs on the surface of the already formed ones [63]. But, as the time passes, the population condition worsens due to depleted nutrients and accumulated metabolic products. Moreover, the BNC gel-film increases in thickness and decreases in air oxygen permeability. Therefore, the length of new, later-synthesized BNC molecules becomes shorter, which is why the averaged degree of polymerization of the sample, as determined by analysis, reduces as well.

The onset temperature of decomposition of the samples ranges from 290.7 °C to 329.3 °C, which is perfectly consistent with the literature data. For instance, it was reported in the study [64] on the thermal stability of different celluloses that this range for BNC was between 184 °C and 379 °C, pointing to an excellent thermal stability.

We consider structural characteristics the most significant measures for BNCs. It is these characteristics that not only help differentiate one BNC type from the other, but also determine the pattern of the hierarchical structure of BNC and, hence, its properties [4,24].

The SEM findings were interpreted in line with [65]. The obtained findings were absolutely atypical. One of the problems that arises when replacing a synthetic nutrient medium by unconventional ones from cellulosic biomass is the reduction in the crystallinity index and Iα cellulose content of BNCs. For example, when the synthetic medium was replaced by a rice husk-derived hydrolyzate for BNC biosynthesis, the BNC index of crystallinity declined to 28% [58], while the use of grape bagasse decreased the Iα allomorph content of BNCs by 72% to 56% [66]. Such a tendency was not detected herein for the microbial producers and nutrient media used, and the transition from the synthetic to hydrolyzate media retained both the crystallinity index and the Iα allomorph content of BNCs. This is probably because the method used herein to pretreat the cellulosic feedstocks provides good-quality nutrient media [25,26].

The predominant content of Iα cellulose allomorph is characteristic of BNC [3]. However, the Iα allomorph content achieved for *Medusomyces gisevii* Sa-12 and *Komagataeibacter xylinus* B-12431 is extremely high: 99.2–100.0%. The Iα allomorph content reported in the literature ranges from 64% [67] to 90% [40].

The BNC samples produced by *Komagataeibacter xylinus* B-12431 and *Medusomyces gisevii* Sa-12 exhibited the highest index of crystallinity between 88% and 100% (as measured in reflection geometry) exceeding all the previously reported values of 46.0% to 95.6% [13,40,68].

## 4. Materials and Methods

### 4.1. Microbial Producers of BNC

Here, we used three microbial producers acquired from the Russian National Collection of Industrial Microorganisms (Scientific Center ‘Kurchatov Institute’, Research Institute for Genetics and Selection of Industrial Microorganisms, Moscow, Russia). The *Komagataeibacter xylinus* B-12429 strain originates from the DSMZ 2004 collection and was isolated by Brown in 1886. The *Komagataeibacter xylinus* B-12431 strain originates from the DSMZ 2325 collection and was isolated by Brown in 1886, with both strains designed for BNC synthesis. The symbiotic culture used herein was the microbial producer *Medusomyces gisevii* Sa-12, also known as Kombucha.

*Medusomyces gisevii* is a symbiotic culture in which acetobacteria and yeast prevail as a microflora. *Medusomyces gisevii* is composed of 8–10 bacterial genera such as *Komagataeibacter*, *Gluconobacter*, *Lyngbya*, *Bifidobacterium*, etc., and 15–30 yeast genera such as *Candida*, *Lachancea*, *Kluyveromyces*, *Zygosaccharomeces*, *Schizosaccharomyces*, etc. [43,45].

### 4.2. Maintenance of Cultures

The Hestrin–Schramm medium, which is classical for microbial producers of BNC, was used to maintain the vital activity of the individual *Komagataeibacter xylinus* strains, and consisted of 2% glucose, 0.5% peptone, 0.5% yeast extract, 0.27% Na_2_HPO_4_, and 0.115% citric acid. The pH was 6.0. To set this pH, HCl and NaOH were used. The medium was sterilized in an autoclave at 0.5 atm for 30 min. The same medium was employed for the control biosynthesis of BNC by the B-12429 and B-12431 strains.

A semisynthetic nutrient medium consisting of glucose and black tea extract was utilized to maintain *Medusomyces gisevii* [37]. This nutrient medium was the control and was prepared as follows: a weighed portion of dry black tea (10 g dry black tea per 1 L nutrient broth, which is equivalent to 3.2 g/L black tea extractives in the medium) was poured over with boiling-hot water, 20 g/L glucose was injected, the extraction vessel was capped, and the contents were maintained at 94–96 °C for 15 min. Afterwards, the mixture was cooled and filtered off. The nutrient medium was not sterilized.

The culture was run under static conditions at 28 ± 0.2 °C for 7 days. The pH was not adjusted during the cultivation, which is optimum for most microbial producers of BNC [13,15,40]. The culture medium was used as the inoculum and injected in the amount of 10%.

### 4.3. Cellulosic Feedstocks and Pretreatment

Miscanthus (*Miscanthus sacchariflorus* (Maxim.) Hack) var. Soranovskiy was selected and grown in the Institute of Cytology and Genetics (Novosibirsk, Russia) [67]. Miscanthus was ground by a Gardena straw grinder (Berlin, Germany) to at most 10 mm. Oat hulls were kindly provided by the ZAO Biysk Elevator (Biysk, Russia).

The feedstocks were pretreated by a two-stage method in a 250-L standard vessel under atmospheric pressure at 90–96 °C at the pilot production site of the IPCET SB RAS. The method involved prehydrolysis with a 1% HNO_3_ solution, treatment with a 4% HNO_3_ solution, water washing, treatment with a 4% NaOH solution and water washing, as described in detail in [25] and briefly in the Appendix A. The chemical compositions of the feedstocks and resultant pulps are outlined in Appendix A.

### 4.4. Enzymatic Hydrolysis of Pulps

The resultant pulps were enzymatically hydrolyzed with the commercial enzymes CelloLux-A and BrewZyme. The initial solid loading was 30 g/L. The enzymatic hydrolysis conditions and results are detailed in [69] and briefly in the Appendix A.

### 4.5. Preparation of Nutrient Media from Enzymatic Hydrolyzates of Pulps

The enzymatic hydrolyzates were centrifuged at 3500 rpm for 10 min; afterwards, they were standardized against glucose via the dilution method to bring the medium to a glucose content of 20 g/L.

The experiments revealed that the individual strains were not capable of producing BNC on the semisynthetic glucose medium supplemented with black tea extractives, on pristine hydrolyzate media, and on hydrolyzate media supplemented with black tea extractives. Therefore, further attempts were made to biosynthesize BNC using the individual strains on hydrolyzate media supplemented with nutrients similarly to the Hestrin–Schramm medium composition (except for glucose). Moreover, for the biosynthesis of BNC via symbiotic *Medusomyces gisevii* Sa-12, the nutrient media were pasteurized at 100 °C with no holding, which turned out to be inappropriate when the individual *Komagataeibacter xylinus* strains were used because of the observed contamination with a foreign microflora. Therefore, the nutrient media were autoclaved at 0.5 atm for 30 min.

While working with the symbiotic culture, the standardized hydrolyzates were heated to 100 °C and employed as extracting agents to recover black tea extractives. No vitamins or growth factors were added.

Prior to biosynthesis, the microbial producers were subjected to adaptation on the enzymatic hydrolyzate media under conditions set forth in Section 4.2.

### 4.6. Biosynthesis of BNC

During the biosynthesis of BNC on enzymatic hydrolyzates from cellulosic feedstocks, the cultivation was performed in 250 mL glass vessels, the nutrient medium volume being 100 mL. Each experiment employed 16 vessels. The inoculum comprised 10% of the nutrient medium volume. The cultivation was carried out at 27 °C for 14 days under static conditions. The pH was not adjusted during the cultivation. Each experiment was run in triplicate. BNC was further purified in line with [25] to furnish semitransparent, pearl-white BNC gel-films.

### 4.7. Calculation of BNC Yield

The yield of BNC was calculated by the equation:(1)$$\eta =\frac{m_{BNC}}{C_{g}\cdot V\cdot 0.9}\cdot 100$$
where *η* is the BNC yield, %; *m_BNC_* is the BNC weight on an oven-dry basis, g; *C_g_* is the glucose concentration of the medium, g/L; *V* is the volume of the medium, L; 0.9 is the recalculation factor attributed to the water molecule detachment upon polymerization of glucose into cellulose [70].

### 4.8. Analytical Techniques

The chemical composition of the cellulosic feedstocks and the resultant pulps (contents of cellulose, hemicelluloses, acid-insoluble lignin and ash) was analyzed according to the reported procedures [71,72,73,74]. The degree of polymerization was measured using a viscometer (OOO Ecroshim, Saint-Petersburg, Russia) using cadoxene (ethylenediamine, AO LenReaktiv, CAS No. 107-15-3, Saint-Petersburg, Russia; cadmium oxide, AO LenReaktiv, CAS No. 1306-19-0, Saint-Petersburg, Russia) according to the procedure described in [75]. The concentration of reducing sugars (RS) calculated as the glucose equivalent was measured using a spectrophotometer (United Products & Instruments, Dayton, NJ, USA) using 3,5-dinitrosalicylic acid (Panreac, CAS No. 609-99-4, Barcelona, Spain), as reported [76]. The content of pentosans calculated in xylose equivalent was measured using a spectrophotometer (United Products & Instruments, Dayton, NJ, USA) using the orcinol–ferric reagent (Acros Organics, a manufacturer of 99% orcinol monohydrate, CAS No. 6153-39-5, Dayton, NJ, USA), as described [77]. The moisture content of the feedstocks and substrates was measured on an Ohaus MB23 moisture analyzer (Ohaus Corporation, Parsippany, NJ, USA). The pH was measured using an I-160 MI ion meter (OOO Izmeritelnaya Tekhnika, Moscow, Russia). The microbial measures (yeast and acetobacterial counts) were monitored using an Optika B-150 microscope (Optika Microscopes, Milan, Italy). Scanning electron microscopy (SEM) of freeze-dried BNC samples was performed using a JSM-840 microscope (JEOL Ltd., Tokyo, Japan) with a Link-860 series II X-ray microanalyzer. Infrared spectroscopy (IR) was performed on an Infralum FT-801 FTIR spectrophotometer (OOO NPF Lumex Sibir, Novosibirsk city, Russia) at 4000–500 cm^−1^ in KBr pellets. Thermogravimetric analysis was performed on a Shimadzu DTG-60 thermogravimetric analyzer (Shimadzu, Kyoto, Japan). The test conditions were as follows: the test sample was heated at a rate of 10 °C/min to a maximum temperature of 600 °C in nitrogen environment at a flow rate of 40 mL/min. The sample weight was *p* = 6.0–6.5 mg. The strength of BNC was measured on a DTG-60 thermomechanical analyzer, whereby the test sample was stretched at a rate of 5.0 g/min from 0.0 g to a maximum load of 400.0 g until failure; the test temperature was 23.0 °C. X-ray diffraction was performed on a DRON-6 monochromatic diffractometer (Burevestnik, Nalchik city, Russia) with Fe-K_α_ radiation at scattering angles of 3° to 145°. Spectral characteristics of the X-ray micrograms were estimated using the PdWin software package v. 4.0. The crystallinity index estimation procedure and the full-profile analysis are described in detail in [35].

This work was performed with equipment provided by the Biysk Regional Center for Shared Use of Scientific Equipment of the SB RAS (IPCET SB RAS, Biysk, Russia).

## 5. Conclusions

Herein, we have investigated for the first time how a microbial producer (*Komagataeibacter xylinus* B-12429 and Komagataeibacter xylinus B-12431 individual strains and symbiotic *Medusomyces gisevii* Sa-12) influences the BNC yield and properties when BNC is produced from cellulosic feedstocks. The cellulosic feedstocks used were miscanthus and oat hulls that were available on a global scale. All the conversion stages were performed under the same conditions and involved pretreatment with dilute nitric acid and sodium hydroxide solutions, enzymatic hydrolysis, standardization of growth media against the contents of reducing sugars and black-tea extractives, and biosynthesis of BNC. The yield of BNC being produced by the symbiotic culture attained 1.8 g/L for oat hulls, which is 44–65% higher than that for the individual strains. For all the microbial producers, the yield of BNC derived from miscanthus was 6–37% lower than that from oat hulls, indicating the presence of inhibitors of BNC biosynthesis in miscanthus itself. The use of the symbiotic culture offers additional technological benefits, as there is no need to add mineral salts and growth factors to the media and it is enough to use pasteurization of the nutrient medium rather than sterilization. The physicochemical properties (nanofibril width, degree of polymerization, elastic modulus, Iα allomorph content, and crystallinity index) of the BNC samples obtained on hydrolyzate media depend, to a greater extent, on the microbial producer type and, to a lesser extent, on the composition of the nutrient medium used. Taken together, the criteria such as BNC yield, processability and BNC index of crystallinity have proved the expedient use of the symbiotic culture for the production of high-value BNC from low-cost cellulosic biomass.

## Figures and Tables

**Figure 1 ijms-24-14401-f001:**
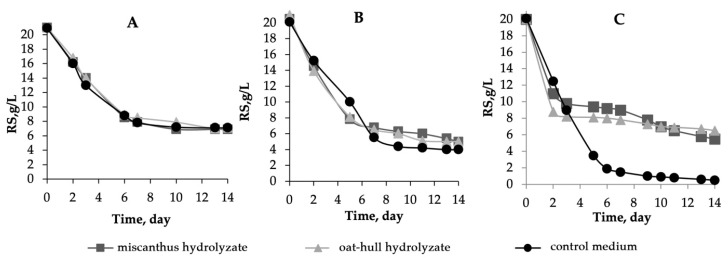
The variation in the RS concentration with time during BNC biosynthesis in miscanthus hydrolyzate medium, in oat-hull hydrolyzate medium, in control medium using of microbial producer: (**A**)—*Komagataeibacter xylinus* B-12429; (**B**)—*Komagataeibacter xylinus* B-12431; and (**C**)—*Medusomyces gisevii* Sa-12. The half-width of the confidence interval for the RS concentration was ±0.2 g/L.

**Figure 2 ijms-24-14401-f002:**
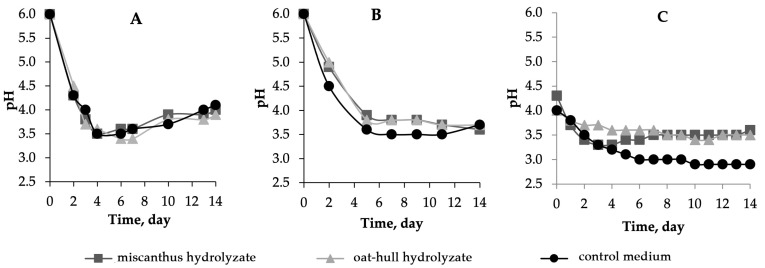
The variation in pH with time during BNC biosynthesis in miscanthus hydrolyzate medium, in oat-hull hydrolyzate medium, in control medium using microbial producer: (**A**)—*Komagataeibacter xylinus* B-12429; (**B**)—*Komagataeibacter xylinus* B-12431; and (**C**)—*Medusomyces gisevii* Sa-12. The half-width of the confidence interval for pH was ±0.1.

**Figure 3 ijms-24-14401-f003:**
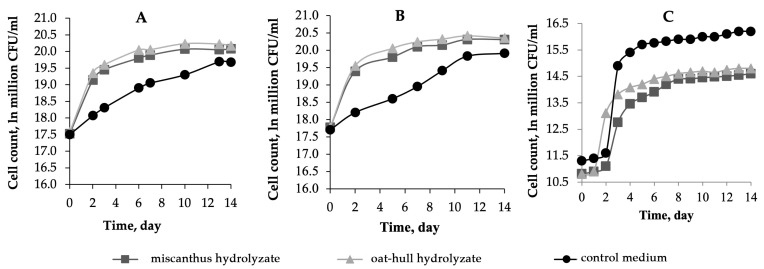
The variation in the acetobacterial count over time during BNC biosynthesis in miscanthus hydrolyzate medium, in oat-hull hydrolyzate medium, in control medium using microbial producer: (**A**)—*Komagataeibacter xylinus* B-12429; (**B**)—*Komagataeibacter xylinus* B-12431; and (**C**)—*Medusomyces gisevii Sa-12.* The half-width of the confidence interval for the cell count was ±0.1 ln million CFU/mL.

**Figure 4 ijms-24-14401-f004:**
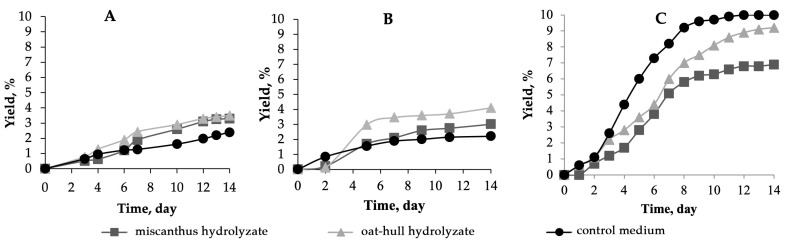
The variation in the BNC yield with time during BNC biosynthesis in miscanthus hydrolyzate medium, in oat-hull hydrolyzate medium, in control medium using microbial producer: (**A**)—*Komagataeibacter xylinus* B-12429; (**B**)—*Komagataeibacter xylinus* B-12431; and (**C**)—*Medusomyces gisevii* Sa-12. The half-width of the confidence interval for the BNC yield was ±0.1%.

**Figure 5 ijms-24-14401-f005:**
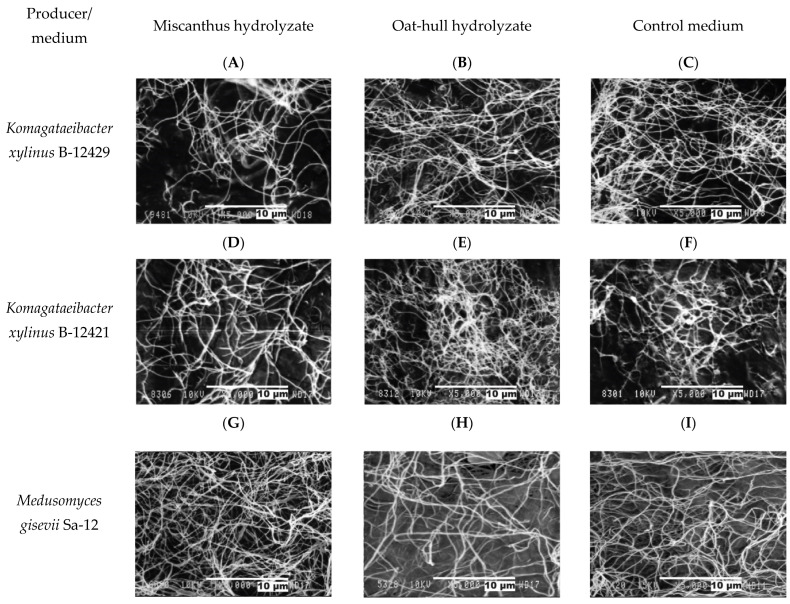
SEM images of BNC samples in 72 h of culture, ×5000 zoom: (**A**–**C**)—*Komagataeibacter xylinus* B-12429; (**D**–**F**)—*Komagataeibacter xylinus* B-12431; (**G**–**I**)—*Medusomyces gisevii* Sa-12; (**A**,**D**,**G**)—miscanthus hydrolyzate; (**B**,**E**,**H**)—oat-hull hydrolyzate; and (**C**,**F**,**I**)—control medium.

**Table 1 ijms-24-14401-t001:** Yield and physicochemical properties of BNC subject to the microbial producer and cellulosic feedstock used compared to the control.

Properties	*Komagataeibacter xylinus* B-12429			*Komagataeibacter xylinus* B-12431			*Medusomyces gisevii* Sa-12		
	MH	OHH	Control	MH	OHH	Control	MH	OHH	Control
BNC yield in 14 days of biosynthesis, % of the RS, ±0.2%	3.3	3.5	2.4	3.0	4.1	1.2	6.3	10.0	10.0
Volumetric productivity of BNC in 14 days of biosynthesis on a dry matter basis, g/L, ±0.1 g/L	0.59	0.63	0.43	0.54	0.74	0.22	1.13	1.80	1.80
Mean width of BNC microfibrils, nm	91.1	90.8	89.6	82.1	90.1	82.8	88.6	82.4	87.0
Degree of polymerization subject to BNC biosynthesis time, ±50									
7 days	880	860	900	2400	2600	1900	1550	1760	2600
14 days	800	820	880	1920	1940	2200	1470	1450	2200
Strength properties									
Breaking strength, MPa, ±0.5 MPa	23.9	41.2	5.7	83.7	65.6	128.3	21.5	26.2	33.2
Conventional yield limit, MPa, ±0.5 MPa	3.9	3.0	1.2	10.9	5.8	8.4	3.0	3.7	5.6
Relative elongation at maximum load, %, ±0.1%	1.4	1.1	0.5	2.6	1.9	2.7	1.2	1.7	1.5
Relative elongation at yield, %, ±0.1%	0.3	0.3	0.1	0.5	0.3	0.4	0.4	0.7	0.6
Sample thickness, µm, ±0.5 µm	10.0	17.5	7.5	5.0	15.0	5.0	30.0	50.0	20.0
Elastic modulus, MPa, ±10.0 MPa	1212	1142	1033	2180	2215	2143	510	529	933
Thermogravimetric data									
Onset temperature of decomposition, °C, ±1.0 °C	305.8	294.8	290.7	301.7	302.1	300.5	317.6	329.3	307.1
Content of cellulose allomorphs and degree of polymerization, as measured via SEM									
Iα allomorph, %, ±1.0%	100.0	100.0	99.2	92.8	94.2	93.5	100.0	100.0	100.0
Iβ allomorph, %, ±1.0%	0.0	0.0	0.8	7.2	5.8	6.5	0.0	0.0	0.0
Crystallinity index (in reflection geometry), %, ±5%	74	87	76	90	91	94	94	93	86
Crystallinity index (in transmission geometry), %, ±5%	75	90	88	100	89	100	92	87	88

Notes: (1) MH—miscanthus hydrolyzate; OHH—oat-hull hydrolyzate; (2) controls were the Hestrin–Schramm medium for *Komagataeibacter xylinus* B-12429 and B-12431 and the control medium supplemented with black tea extract for *Medusomyces gisevii* Sa-12; (3) some data on BNCs synthesized by *Medusomyces gisevii* Sa-12 were reported previously [25,26].

**Table 2 ijms-24-14401-t002:** Comparison of BNC biosynthesis efficiency between individual strains and symbiotic culture on hydrolyzate media.

Measure	Individual Strains	Symbiotic Culture
Nutrient salts added	Yes	No
Tea extractives added	No	Yes
Sterilized medium	Yes	No
Microbial control of BNC production	Easy and automatable	Sophisticated and cannot be automated
BNC yield on hydrolyzate media compared to that on synthetic one	38–46% higher	Up to 37% lower
BNC yield on hydrolyzate media, %	3.0–4.1	6.3–10.0

## Data Availability

Not applicable.

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
