# Peer review of "Biosynthesis of Bacterial Nanocellulose from Low-Cost Cellulosic Feedstocks: Effect of Microbial Producer"

_ijms, 2023, doi:10.3390/ijms241814401_

Round 1

Reviewer 1 Report

The manuscript submitted by Skiba et al. mainly focused on the effects of microbial producers on BC production from low-cost cellulosic feedstocks. They said the main aim of this work was to answer the questions addressed by reviewers during their last paper’s peer-review process. To my opinion, I don’t think this concern can be an important perspective. Although the work was well-performed and the story was fluent, it doesn’t sound important enough. Another question is that the work didn’t do any molecular experiment, so it seems that the MS cannot fall within the scope of IJMS.

1.     The 1st question I concerned is that the authors have claimed that a low-cost or green strategy should be taken into account to produce more biodegradable polymers in the introduction. However, the feedstocks they used to produce BC is still needed to be pretreated by a chemical or enzymatic hydrolysis. I notice that there are several low-cost feedstocks have been tried to prepare BC without any pretreatment, which were detailed in https://doi.org/10.1007/s10570-019-02868-1. So, the novelty background given in the introduction should be updated.

2.     Line 105. As shown in Fig. 2C, why the initial pH in Medusomyces gisevii Sa-12 system was significantly lower than that in individual systems? They are not comparable.

3.     Line 112. Pls give us the data rather than state a confuse result. We cannot catch the data by our naked eyes.

4.     Line 139. Pls replace “i.e. [29] for the В-12429 strain and [30] for the В-12431 strain” with “i.e. for the В-12429 strain [29] and for the В-12431 strain [30]”.

5.     Line 149. “between 82 nm to 91 nm”, you didn’t analyze the diameter of nanofibers, why did you get this data? As calculated by software, the diameter of BC nanofiber is about 40-50 nm. If you got the true data, pls also update Table 1.

Author Response

The authors' response to the reviewer's comments has been uploaded.

Reviewer 2 Report

The authors present the research work entitled: Biosynthesis of bacterial nanocellulose from low-cost cellulosic feedstocks: effect of microbial producer. This study is interesting and of great importance for the use of the symbiotic culture in transforming low-cost feedstocks into highly crystalline BNC. 

Line 9 (Abstract): The conclusions must be written based on the general objective and present only the most relevant results. The aim should be incorporated into the abstract.

Line 16-17: The yield of  bacterial nanocellulose (BNC)  should be shown in mg/L or g/L to compare with the production of other existing works, it should also be considered if it is on a dry or wet basis.

Fig. 1, 2, 3: The graphs do not present replicates or standard deviations. The yields of BNC are very low and there is not a good discussion of the results. It is necessary to compare and improve the discussion of the results. Not only with the synthetic culture, but with other studies.

Line 418: There is a lack of information on the methodology. For example, on line 418: What was the weight of the sample? What type of atmosphere was used in the analysis? 

The conclusions are not displayed and are not clear. The conclusions must be written based on the general objective and present only the most relevant results.

They do not present the design of experiments. The graphs do not present standard deviation or statistical analysis

There are many references that are not necessary.

Author Response

(The authors gave the same response as above.)

Reviewer 3 Report

1.      In the introductory section, describe the existing cellulosic feedstock and provide economic (low-cost) evidence for the feedstock selected in this study.

2.      In the result section (2.1), explain the reason for the results and cite the appropriate references.

Author Response

(The authors gave the same response as above.)

Reviewer 4 Report

The author did an interesting work. I have read the article carefully and I have the following comments - 

1. The figures 1-4 needs to be improved. All the figures should have appropriate axis title and legends. It is very tough to follow otherwise and makes no sense. 

2. In the SEM, figure 5, scale bar should be added into the figure. 

3. It is really hard to follow table 1. As I tried to understand strength properties and thermogravimetric data, I could not keep track and connect the data. 

4. Also, the authors should summarize the finding in conclusion. It is very hard to understand whether the aim of the article was achieved without any conclusion. 

English could be improved.

Author Response

(The authors gave the same response as above.)

Round 2

Reviewer 1 Report

The authors have revised the MS according to reviewers' suggestions. I recommend its potential in IJMS as is. By the way, the authors can calculate the nano diameter by Nano Measurer 1.2 in the future work.

Author Response

Dear Reviewer 1!

Thank you for recommending the use of the program and supporting our article!

Reviewer 2 Report

The presentation of the data has not changed, there is no statistical analysis or design of experiments that supports the work. The graphs have no standard deviation.

Line 16-17: The yield of  bacterial nanocellulose (BNC)  should be shown in mg/L or g/L to compare with the production of other existing works, it should also be considered if it is on a dry or wet basis.

Fig. 1, 2, 3: The graphs do not present replicates or standard deviations. The yields of BNC are very low and there is not a good discussion of the results. It is necessary to compare and improve the discussion of the results, not only with the synthetic culture media but with other studies.

Line 418: There is a lack of information on the methodology. For example, on line 418: What was the weight of the sample? What type of atmosphere was used in the analysis? 

The conclusions are not displayed and are not clear. The conclusions must be written based on the general objective and present only the most relevant results.

They do not present the design of experiments. The graphs do not present standard deviation or statistical analysis

Author Response

Dear Reviewer 2!

We apologize. A technical failure has occurred. Our detailed responses to your valuable comments can be found in the attached file. The corrected article, taking into account your comments, is available to you after the first round.

Reviewer 4 Report

The manuscript still need more work. The Table 1 is very clumsy. It is really hard to understand the significance of data without any standard deviation. 

Need to improve the english.

Author Response

Dear Reviewer 4!

We apologize. A technical failure has occurred. Our detailed responses to your valuable comments can be found in the attached file. The corrected article, taking into account your comments, is available to you after the first round. 

The table is really very cumbersome, but all the data is in one place and it is easy to compare them. We have added standard deviations.
